# A CFOA-Based Voltage-Mode Multifunction Biquadratic Filter and a Quadrature Oscillator Using the CFOA-Based Biquadratic Filter

**San-Fu Wang [1], Hua-Pin Chen [2,*] , Yitsen Ku [3] and Po-Yu Chen [2]**

[1]    Department of Electronic Engineering, National Chin-Yi University of Technology, Taiping,
     Taichung 41170, Taiwan; sf_wang@ncut.edu.tw
[2]    Department of Electronic Engineering, Ming Chi University of Technology, New Taipei 24301, Taiwan;
     m06158019@o365.mcut.edu.tw
[3]    Department of Electrical Engineering, California State University Fullerton, Fullerton,
     California, CA 92831, USA; joshuaku@fullerton.edu
*    Correspondence: hpchen@mail.mcut.edu.tw; Tel.: +886-22-908-9899; Fax: +886-22-908-5247

**Abstract:** This paper proposes a new high-input impedance current feedback operational amplifier (CFOA)- based voltage-mode multifunction biquadratic filter and a voltage-mode quadrature oscillator using the proposed high-input impedance CFOA-based biquadratic filter. The proposed high-input impedance CFOA-based voltage-mode multifunction biquadratic filter uses three CFOAs, three resistors, and two grounded capacitors with two inputs and three outputs. The filter can simultaneously realize non-inverting low-pass, non-inverting band-pass, and non-inverting band-reject filtering functions at the high-input impedance terminal while the inverting band-pass and non-inverting high-pass filtering functions can also be obtained by applying another high-input impedance terminal. The filter offers orthogonal control of resonance angular frequency and quality factor. The proposed high-input impedance CFOA-based voltage-mode multifunction biquadratic filter can be used to implement a voltage-mode quadrature oscillator with an independently controlled the frequency of oscillator and the condition of oscillation. The OrCAD PSpice simulation and experimental results of the commercially available integrated circuit, AD844AN, are used to confirm the characteristics of the proposed filter and oscillator.

**Keywords:** voltage-mode filter; quadrature oscillator; high-input impedance; CFOA

## 1. Introduction

Active filters and oscillators have received significant attention in analog systems because they are frequently used in electrical and electronic engineering works [1–7]. The current feedback operational amplifiers (CFOAs) are very attractive elements used for realizing active filters and oscillators. CFOAs have an extended bandwidths and higher slew rates advantage over conventional voltage operational amplifiers [8–10]. The CFOA has terminal characteristics like positive second generation current-conveyor (CCII+) followed by a voltage buffer (VB) [11,12]. The most commercial CFOA, such as the AD844AN from Analog Devices, has obtained the acceptance of researchers as an active building block in analog circuit design [13,14]. The CFOA differs from the traditional operational amplifier in that the voltage on the non-inverting input terminal is transferred to the inverting input terminal and the inverting input terminal is of low input impedance.

Many voltage-mode CFOA-based universal biquadratic filters have been proposed in the literature [14–20]. These circuit configurations can realize all the standard transfer functions by selecting different input voltages. However, they require the injection of an excitation signal from

one terminal of the capacitor in each circuit design and only one/two standard filtering functions at most can be simultaneously obtained in each circuit realization. A multiple inputs and single output universal filter using four CFOAs, five resistors, and two grounded capacitors has been reported in the literature [21]. This circuit provides the features of orthogonal control of resonance angular frequency ($\omega_o$) and quality factor (Q), using only grounded capacitors as well as high-input impedance. However, this reported circuit needs to employ four CFOAs and needs a component matching condition for realizing the all-pass filtering function. The configuration in the literature [22] proposed another three-input single-output CFOA-based biquadratic filter. This configuration employs three CFOAs, five resistors, and two grounded capacitors. The circuit also provides the features of orthogonal control of $\omega_o$ and Q, using only grounded capacitors as well as high-input impedance. However, this circuit still needs a component matching condition for realizing the all-pass filtering function. It is known that a useful feature of the multifunction biquadratic filter is that the low-pass, band-pass, high-pass, and/or band-reject outputs are simultaneously available at various nodes in the circuit. These additional outputs can be used in some systems that employ more than one filter function [23]. Hence, four CFOA-based universal biquadratic filters were proposed in the literature [24]. These proposed configurations can simultaneously realize low-pass, band-pass, and high-pass responses and have several other possible applications such as in crossover networks used in three-way high-fidelity loudspeakers, and phase-locked loop frequency modulation stereo demodulators [25]. Of particular interest here are the voltage-mode CFOA-based multifunction biquadratic filters [26–29]. Therefore, a voltage-mode non-inverting low-pass (LP), inverting band-pass (IBP), and non-inverting band-reject (BR) filter employing three CFOAs, three resistors, and two grounded capacitors has been proposed in the literature [26] but it does not offer high-input impedance. Another voltage-mode LP, non-inverting band-pass (BP), and non-inverting high-pass (HP) filter employing three CFOAs, four resistors, and three grounded capacitors has been proposed in the literature [27] but it suffers from the drawback of using three capacitors. A high-input impedance voltage-mode LP, IBP, HP, and BR filter employing four CFOAs, four resistors, and two grounded capacitors has been proposed in the literature [28] but it suffers from the drawback of using four CFOAs. A high-input impedance voltage-mode LP, BP, and HP filter employing three CFOAs, four resistors, and two grounded capacitors has been proposed in the literature [29] but it has a capacitor connected in series to the X port of the CFOA. Because the CFOA has a non-negligible output parasitic resistance on port X, $R_X$, when the X port of CFOA is loaded by a capacitor, it leads to an improper transfer of functions.

In this paper, a new high-input impedance multifunction biquadratic filter configuration with two inputs and three outputs using three CFOAs, three resistors, and two grounded capacitors has been presented. The circuit realizes LP, BP, and BR voltage responses, simultaneously, by the use of only a single voltage input signal while the IBP and HP responses can be obtained by the use of another high-input impedance of input voltage signal. The circuit exhibits high-input impedance and orthogonal control of $\omega_o$ and Q. Moreover, the proposed CFOA-based voltage-mode multifunction biquadratic filter can be easily converted to a voltage-mode quadrature oscillator. Contrary to the previously reported CFOA-based voltage-mode multifunction biquadratic filters [26–29], the proposed filter meets all of the following advantages—(i) employs only three CFOAs, three resistors, and two grounded capacitors, (ii) can have simultaneously LP, BP, and BR filtering responses with a single input and three outputs by the use of only a single voltage input signal while the IBP and HP filtering responses can be realized by the use of another input voltage signal, (iii) has high-input impedance, (iv) has orthogonal control of $\omega_o$ and Q, (v) has no capacitors bringing extra poles that degrade its high frequency performance, (vi) requires no inverting-type voltage input signals, and (vii) can transfer into a voltage-mode quadrature sinusoidal oscillator easily. The properties of the proposed filter and oscillator have been verified by PSpice simulations and furthermore by experimental measurements. The proposed circuit is compared with previously published voltage-mode multifunction biquadratic filters using CFOAs in Table 1.

**Table 1.** Comparison of the proposed circuit with previously CFOA-based multifunction biquadratic filters in the literature [26–29].

| Parameter | (i) | (ii) | (iii) | (iv) | (v) | (vi) | (vii) |
|---|---|---|---|---|---|---|---|
| Reference [26] | yes | no | no | yes | yes | yes | no |
| Reference [27] | no | no | yes | yes | yes | yes | no |
| Reference [28] | no | yes | yes | yes | yes | yes | no |
| Reference [29] | no | no | yes | no | no | yes | no |
| Proposed | yes | yes | yes | yes | yes | yes | yes |

## 2. Proposed Circuits

### 2.1. Voltage-Mode Multifunction Biquadratic Filter

The CFOA is a four-terminal analog building block shown symbolically in Figure 1 [11], with a describing matrix equation of the following form:

$$\begin{bmatrix} I_Y \\ V_X \\ I_Z \\ V_O \end{bmatrix} = \begin{bmatrix} 0 & 0 & 0 & 0 \\ 1 & 0 & 0 & 0 \\ 0 & 1 & 0 & 0 \\ 0 & 0 & 1 & 0 \end{bmatrix} \begin{bmatrix} V_Y \\ I_X \\ V_Z \\ I_O \end{bmatrix} \tag{1}$$

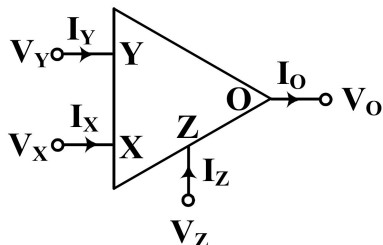

**Figure 1.** Schematic symbol of CFOA.

The CFOA is realized by the cascaded connection of a CCII+ followed by a VB, as shown in Figure 2 [11,12]. It gathers the advantages of the CCII+ and VB such as—high input impedance (Y terminal), low input impedance (X terminal), low output impedance (O terminal), high output impedance (Z terminal), high accuracy, and large bandwidth. The voltage at terminal X follows the voltage at terminal Y in magnitude. The output current at terminal Z follows the current at terminal X in magnitude. The voltage drop at terminal Z will be send to terminal O with the unity voltage gain. The proposed high-input impedance voltage-mode CFOA-based biquadratic filter with two inputs and three outputs is shown in Figure 3, which consists of three CFOAs, two grounded capacitors, and three resistors. The use of the grounded capacitors is particularly attractive for integrated circuit (IC) implementation. In Figure 3, the input voltage signal is connected directly to the Y-terminal of the CFOA and the input current to the Y-terminal of the CFOA is zero, the circuit has the feature of high-input impedance. Routine analysis of the filter in Figure 3 gives the following three output voltages:

$$V_{o1} = \frac{V_{i1} - sC_2R_2V_{i2}}{s^2C_1C_2R_2R_3 + sC_1R_1 + 1} \tag{2}$$

$$V_{o2} = \frac{sC_1R_3V_{i1} + (sC_1R_1 + 1)V_{i2}}{s^2C_1C_2R_2R_3 + sC_1R_1 + 1} \tag{3}$$

$$V_{o3} = \frac{(s^2C_1C_2R_2R_3 + 1)V_{i1} + s^2C_1C_2R_1R_2V_{i2}}{s^2C_1C_2R_2R_3 + sC_1R_1 + 1} \tag{4}$$

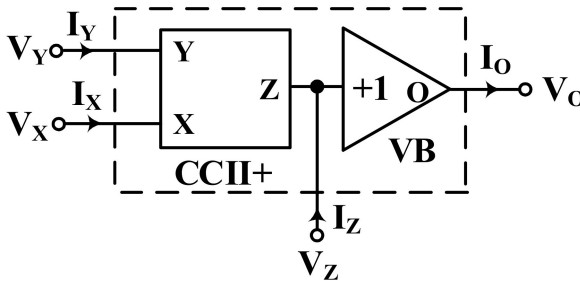

**Figure 2.** Implementation of CFOA given by a CCII+ and a VB.

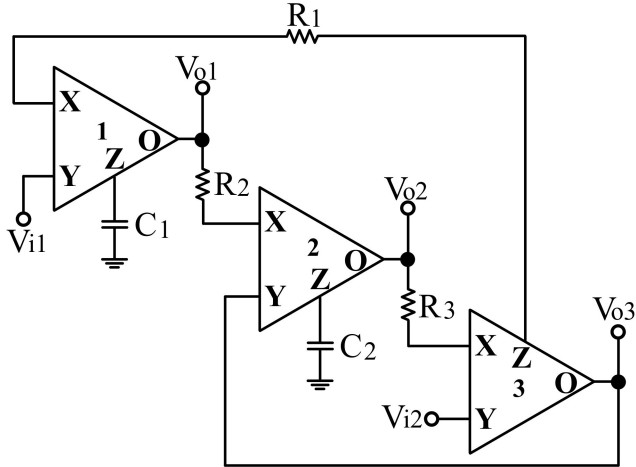

**Figure 3.** The proposed high-input impedance CFOA-based multifunction biquadratic filter.

As indicated by Equations (2)–(4), if $V_{i2} = 0$ (grounded) and $V_{i1} = V_{in}$ is given by the input voltage signal then the LP, BP, and BR filtering functions are given by:

$$\frac{V_{o1}}{V_{in}} = \frac{V_{LP}}{V_{in}} = \frac{1}{s^2 C_1 C_2 R_2 R_3 + s C_1 R_1 + 1} \tag{5}$$

$$\frac{V_{o2}}{V_{in}} = \frac{V_{BP}}{V_{in}} = \frac{s C_1 R_3}{s^2 C_1 C_2 R_2 R_3 + s C_1 R_1 + 1} \tag{6}$$

$$\frac{V_{o3}}{V_{in}} = \frac{V_{BR}}{V_{in}} = \frac{s^2 C_1 C_2 R_2 R_3 + 1}{s^2 C_1 C_2 R_2 R_3 + s C_1 R_1 + 1} \tag{7}$$

By letting $V_{i1} = 0$ (grounded) and proving $V_{i2} = V_{in}$, the IBP and HP filtering functions can be easily obtained from the two voltage outputs $V_{o1}$ and $V_{o3}$ as follows:

$$\frac{V_{o1}}{V_{in}} = \frac{V_{IBP}}{V_{in}} = \frac{-s C_2 R_2}{s^2 C_1 C_2 R_2 R_3 + s C_1 R_1 + 1} \tag{8}$$

$$\frac{V_{o3}}{V_{in}} = \frac{V_{HP}}{V_{in}} = \frac{s^2 C_1 C_2 R_1 R_2}{s^2 C_1 C_2 R_2 R_3 + s C_1 R_1 + 1} \tag{9}$$

As illustrated in Equations (5)–(9), the proposed filter simultaneously provides LP, BP, and BR voltage responses while the IBP and HP responses can also be obtained. The pass-band gains, $G_{BP}$, $G_{IBP}$, and $G_{HP}$, for the BP, IBP, and HP responses are given by:

$$G_{BP} = \frac{R_3}{R_1}, \ G_{IBP} = -\frac{C_2 R_2}{C_1 R_1}, \ G_{HP} = \frac{R_1}{R_3} \tag{10}$$

Table 2 summarizes the two inputs and three outputs CFOA-based voltage-mode multifunction biquadratic filter of the transfer functions.

**Table 2.** Two inputs and three outputs filtering functions realized.

| Input Conditions | Output Terminals | Filter Functions | Passband Gains |
|---|---|---|---|
| Only $V_{i1}$ as input signal | $V_{o1}$ | non-inverting low-pass | unity |
| | $V_{o2}$ | non-inverting band-pass | $\frac{R_3}{R_1}$ |
| | $V_{o3}$ | non-inverting band-reject | unity |
| Only $V_{i2}$ as input signal | $V_{o1}$ | inverting band-pass | $-\frac{C_2 R_2}{C_1 R_1}$ |
| | $V_{o2}$ | none | none |
| | $V_{o3}$ | non-inverting high-pass | $\frac{R_1}{R_3}$ |

Comparing the expression for the denominator to the characteristic equation $D(s) = s^2 + \frac{\omega_0}{Q}s + \omega_0^2$, the parameters $\omega_o$ and Q of the proposed filter are given by:

$$\omega_o = \sqrt{\frac{1}{C_1 C_2 R_2 R_3}} \tag{11}$$

$$Q = \frac{1}{R_1}\sqrt{\frac{C_2 R_2 R_3}{C_1}} \tag{12}$$

Equations (11) and (12) show that the parameter Q can be tuned by adjusting the resistor $R_1$ without affecting $\omega_o$. In other words, Q and $\omega_o$ are orthogonally adjustable. The orthogonal controllability is a desirable property for the designing and tuning flexibility of the biquadratic filters.

### 2.2. Voltage-Mode Quadrature Sinusoidal Oscillator

In Figure 3, by inserting a resistance between the voltage output $V_{o1}$ and terminal X of the third CFOA, the presented CFOA-based voltage-mode multifunction biquadratic filter can be converted into a voltage-mode quadrature oscillator by making $V_{i1} = V_{i2} = 0$. As shown in Figure 4, a new voltage-mode quadrature sinusoidal oscillator based on the proposed filter in Figure 3 is proposed.

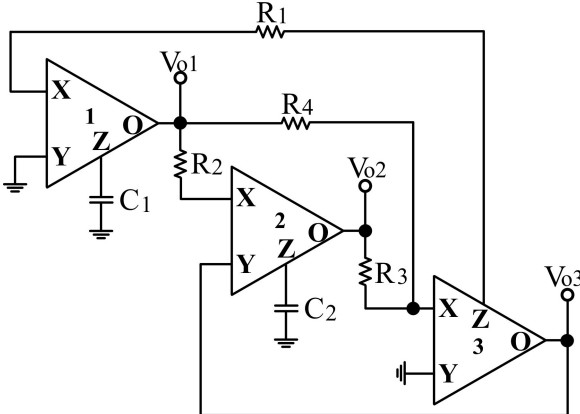

**Figure 4.** The proposed quadrature sinusoidal oscillator using the filter in Figure 3.

The characteristic equation of the quadrature oscillator in Figure 4 can be expressed as:

$$s^2 C_1 C_2 R_2 R_3 + s(C_1 R_1 - C_2 \frac{R_2 R_3}{R_4}) + 1 = 0 \tag{13}$$

The frequency of oscillator (FO) and the condition of oscillation (CO) of Figure 4 can be obtained as:

$$\text{FO}: \ \omega_\text{o} = \sqrt{\frac{1}{C_1 C_2 R_2 R_3}} \tag{14}$$

$$\text{CO}: \ C_1 R_1 R_4 \leq C_2 R_2 R_3 \tag{15}$$

For $R_1 = R_2 = R$, and $C_1 = C_2 = C$, Equations (14) and (15) simplifies to:

$$\text{FO}: \ \omega_\text{o} = \frac{1}{C} \sqrt{\frac{1}{RR_3}} \tag{16}$$

$$\text{CO}: \ R_4 \leq R_3 \tag{17}$$

From Equations (16) and (17), the FO can be controlled by adjusting the value of resistor R without affecting the CO. The CO can be controlled by adjusting the value of resistor $R_4$ without affecting the FO. This means that the FO and CO can be controlled independently by different two resistors. The two marked voltages, $V_{o1}$ and $V_{o3}$, in the circuit of Figure 4 are related as:

$$\frac{V_{o1}}{V_{o3}} = -\frac{1}{sC_1 R_1} \tag{18}$$

Under sinusoidal steady state, Equation (18) can be expressed as:

$$\frac{V_{o1}}{V_{o3}} = \frac{1}{\omega_\text{o} C_1 R_1} e^{j\varphi} \tag{19}$$

where the phase difference is of $\varphi = 90°$, ensuring the voltages, $V_{o1}$ and $V_{o3}$, are in quadrature.

## 3. Non-Idealities Analysis

Taking the non-idealities of the CFOA into account, Figure 5 shows the non-ideal behavior of a CFOA that has the parasitic input resistance, $R_X$, at the X-terminal. The parasitic resistances, $R_Y$ and $R_Z$, and parasitic capacitances, $C_Y$ and $C_Z$, appear between the high-impedance Y and Z terminals and ground, respectively [22]. In the presence of these parasitic elements, the proposed CFOA-based biquadratic filter as shown in Figure 3 can be modified to Figure 6. In Figure 6, the influence of the parasitic resistance, $R_X$, and capacitance, $C_Z$, can be reduced, by selecting the external resistor and capacitor to be much greater than the parasitic resistance and capacitance, respectively. Hence, the effect of parasitic resistance can be easily absorbed as a part of the main resistance and the effect of parasitic capacitance can be easily absorbed as a part of the main capacitance. To reduce the parasitic impedance effects of CFOAs at the Z terminals, the following conditions must be satisfied:

$$\frac{1}{s(C_1 + C_{Z1})} << R_{Z1} \tag{20}$$

$$\frac{1}{s(C_2 + C_{Z2})} << R_{Z2} \tag{21}$$

$$R_1 + R_{X1} << \frac{R_{Z3}}{1 + sR_{Z3} C_{Z3}}. \tag{22}$$

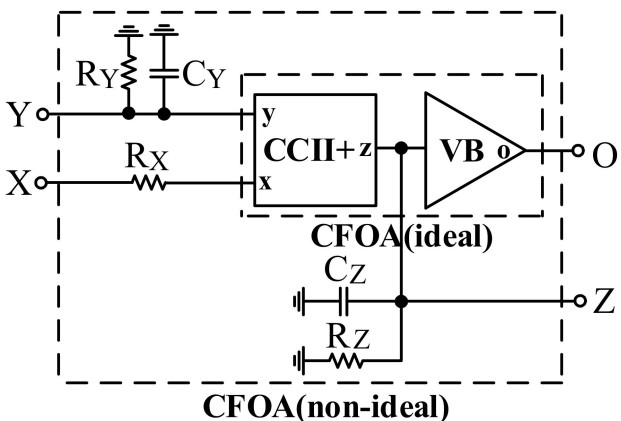

**Figure 5.** General equivalent circuit of the non-ideal CFOA.

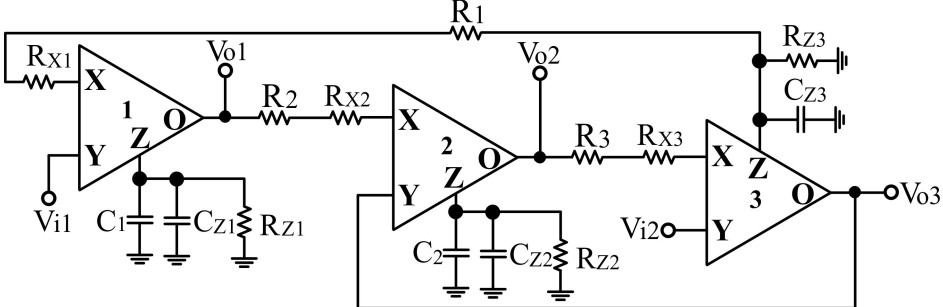

**Figure 6.** Proposed voltage-mode biquadratic filter including the parasitic elements of CFOA.

## 4. Simulation and Experimental Results

### 4.1. Voltage-Mode Multifunction Biquadratic Filter

The proposed CFOA-based filer was simulated using Cadence OrCAD PSpice (version 16.6, San Jose, CA, USA), and measured using the keysight E5061B-3L5 network analyzer (Santa Rosa, CA, USA), and Agilent N9000A CXA signal analyzer (Santa Rosa, CA, USA). The CFOAs were realized using the OrCAD PSpice macro-model of the AD844AN by Analog Devices as well as implemented in hardware using IC AD844AN [30]. The AD844AN supply voltages and capacitor values were selected as $V_{DD} = -V_{SS} = 6V$ and $C_1 = C_2 = 1$ nF. Figures 7–11 show the simulated and measured filtering responses with the equal resistors of $R_1 = R_2 = R_3 = 4$ k$\Omega$ for Q = 1 and $f_o = 39.79$ kHz, respectively. Figure 12 shows the variation in the angular frequency without affecting the quality factor. In Figure 12, equal resistors with $R_1 = R_2 = R_3$ have different values of 4, 2, and 1 k$\Omega$, resulting in Q = 1 for the HP filter and angular frequencies of $f_o = 39.79$, 79.58, and 159.16 kHz, respectively. Next, the quality factor changes without affecting the angular frequency, as shown in Figure 13. This design is for an angular frequency of $f_o = 39.79$ kHz with $R_2 = R_3 = 4$ k$\Omega$ and only varying the value of $R_1$ as 5.6, 4, and 1.6 k$\Omega$, resulting in Q = 0.71, 1, and 2.5 for the BP filter, respectively.

To test the input dynamic range of the proposed filter, the simulation of the IBP filter as an example was repeated for a sinusoidal input signal with $f_o = 39.79$ kHz. Figure 14 shows the time-domain of the IBP response with $V_{i1} = 0$ and $V_{i2} = V_{in}$ at the $V_{o1}$ output terminal, where all the resistor values are 4 k$\Omega$. In Figure 14, the percentage of total harmonic distortion (THD) is 0.6%, when the sinusoidal input signal of an amplitude 1 $V_P$ at 39.79 kHz was applied. The dependence of the IBP filter output harmonic distortion on the input signal amplitude is shown in Figure 15. As can be seen in Figure 15, when the input signal amplitude increases by 2 $V_p$, its total harmonic distortion will increase by 3.2%. The intermodulation distortion (IMD) of the IBP response was studied. Figure 16 shows the dependence of the third-order IMD of IBP filter using two closely spaced tones, $f_1 = 38.79$ kHz and $f_2 = 40.79$ kHz,

with equal input signal amplitude. The proposed filter was investigated using Monte-Carlo analysis. All the passive component deviations in the proposed filter were investigated using Monte-Carlo analysis. The Monte-Carlo simulations were performed on 100 samples. The Gaussian variations of all passive elements were 5%. Figure 17 shows a histogram of the angular frequencies obtained from the BP response Monte-Carlo analysis.

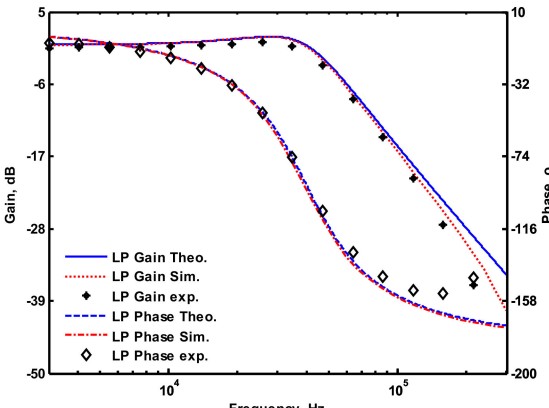

**Figure 7.** Simulation and experimental results of LP filtering responses at $V_{o1}$ when $V_{i1} = V_{in}$ and $V_{i2} = 0$.

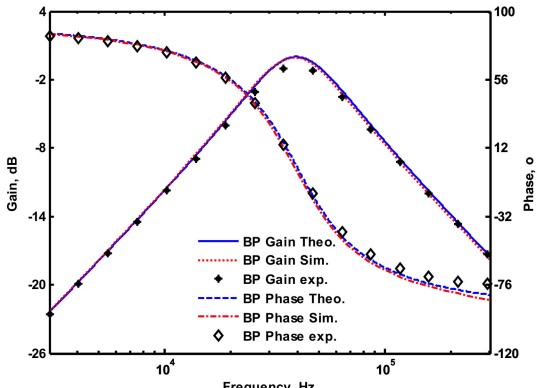

**Figure 8.** Simulation and experimental results of BP filtering responses at $V_{o2}$ when $V_{i1} = V_{in}$ and $V_{i2} = 0$.

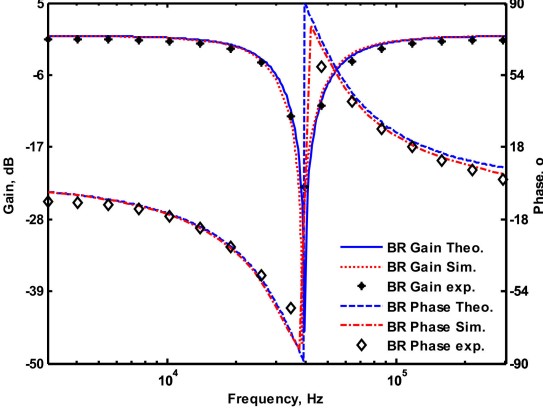

**Figure 9.** Simulation and experimental results of BR filtering responses at $V_{o3}$ when $V_{i1} = V_{in}$ and $V_{i2} = 0$.

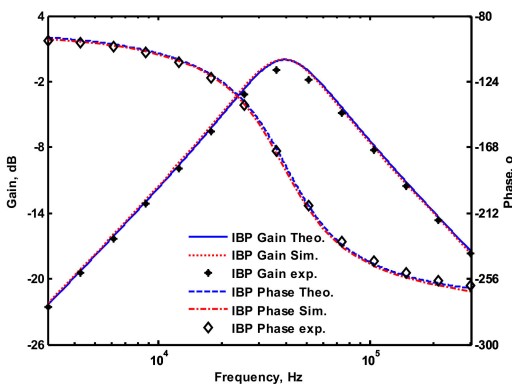

**Figure 10.** Simulation and experimental results of IBP filtering responses at $V_{o1}$ when $V_{i1} = 0$ and $V_{i2} = V_{in}$.

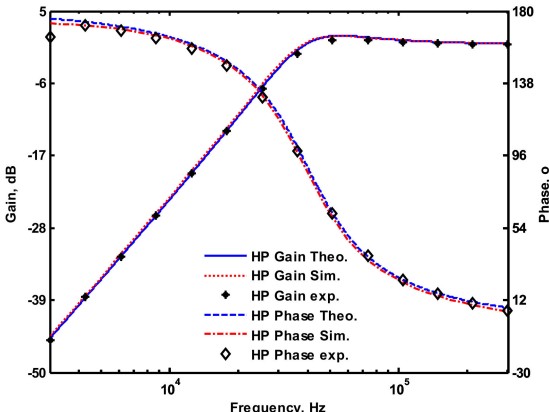

**Figure 11.** Simulation and experimental results of HP filtering responses at $V_{o3}$ when $V_{i1} = 0$ and $V_{i2} = V_{in}$.

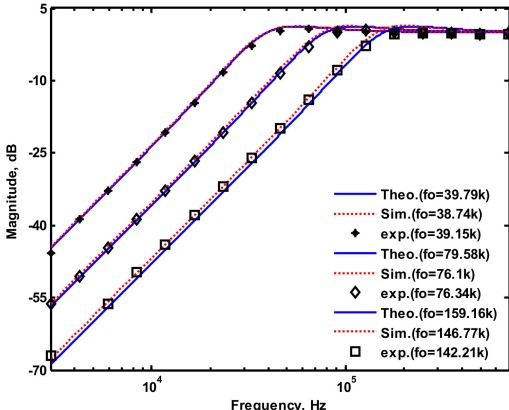

**Figure 12.** Simulation and experimental results of HP filtering responses at $V_{o3}$ when $V_{i1} = 0$ and $V_{i2} = V_{in}$.

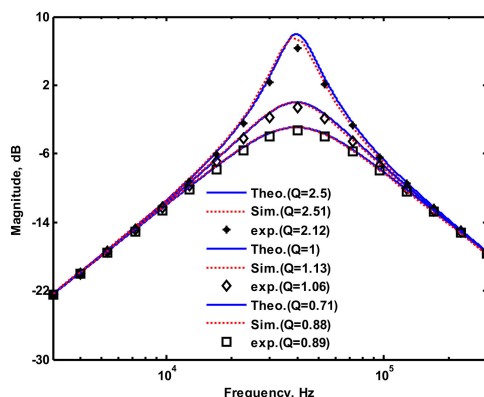

**Figure 13.** Simulation and experimental results of BP filtering responses at $V_{o2}$ when $V_{i1} = V_{in}$ and $V_{i2} = 0$.

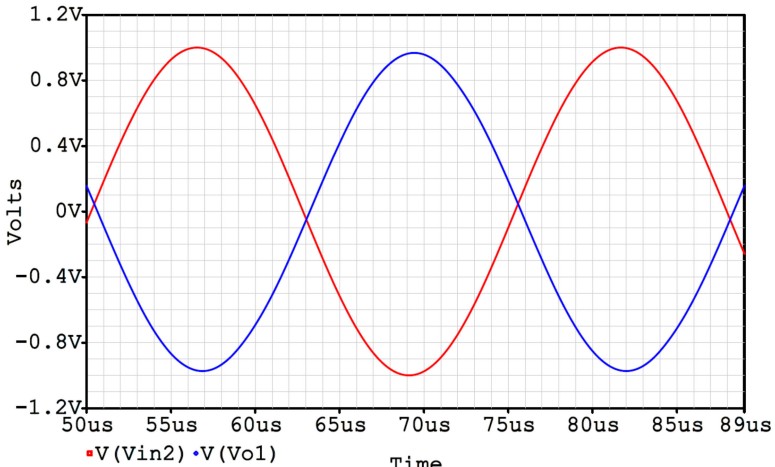

**Figure 14.** Time domain response of the IBP filter at $V_{o1}$ when $V_{i1} = 0$ and $V_{i2} = V_{in}$.

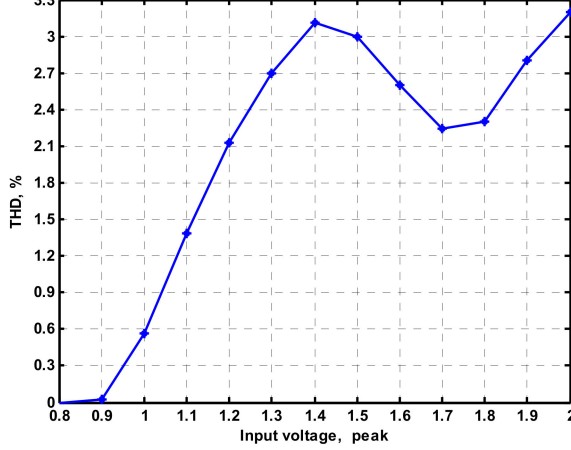

**Figure 15.** Dependence of output harmonic distortion of the IBP filter on an input signal amplitude.

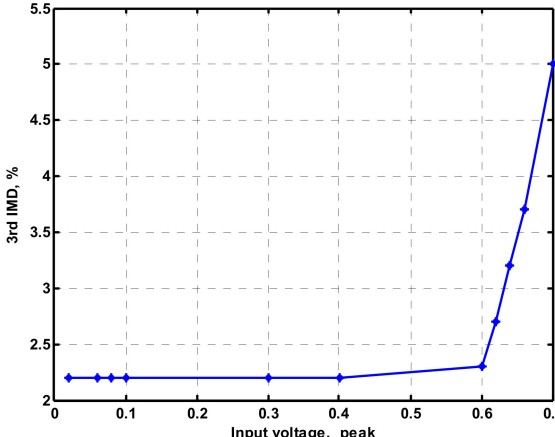

**Figure 16.** Dependence of the third-order IMD of the IBP filter on input signal amplitudes.

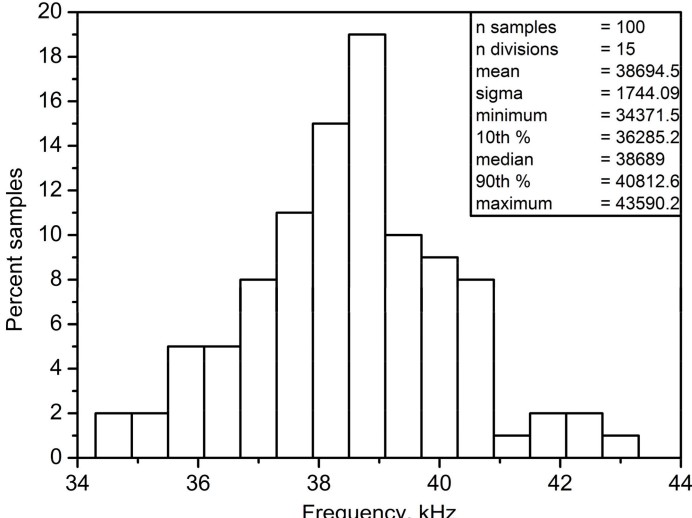

**Figure 17.** Histogram of Monte-Carlo sensitivity analysis of BP filter at $V_{o2}$ when $V_{i1} = V_{in}$ and $V_{i2} = 0$.

To demonstrate the validity of the theoretical analysis, the gain and phase responses obtained through the keysight E5061B-3L5 network analyzer are depicted in Figures 18–22, respectively. In Figures 18–22, the theoretical quality factor was Q = 1 and theoretical angular frequency was $f_o$ = 39.79 kHz, when the passive elements were set as $C_1 = C_2$ = 1 nF and $R_1 = R_2 = R_3$ = 4 kΩ. The power consumption was approximately 180 mW under ± 6 V of supply voltages and 15 mA constant output current. In Figure 23, the measured values of angular frequency were equal to 39.15 kHz, 76.34 kHz, and 142.21 kHz, respectively, by adopting $C_1 = C_2$ = 1 nF and equal resistors of $R_1 = R_2 = R_3$ with 4 kΩ, 2 kΩ, 1 kΩ. Selecting $R_2 = R_3$ = 4 kΩ and different values for $R_1$ as 5.6, 4, and 1.6 kΩ, results in BP filters with theoretical angular frequency $f_o$ = 39.79 kHz and quality factors of Q = 0.71, 1, and 2.5, respectively, as shown in Figure 24. The measured values of Q were equal to 0.89, 1.06, and 2.12, respectively.

To represent the linearity of proposed filter, the 1 dB power gain compression point (P1dB) is measured through the Agilent N9000A CXA signal analyzer. Figure 25 shows the measured P1dB of the IBP filter with input power at the angular frequency of 39.79 kHz. As shown in Figure 25, the measured P1dB of the IBP filter at $V_{o1}$ is about 12 dBm with respect to input power. Figure 26 shows the spectrum of the IBP filter through inter-modulation characterization by applying two tone signals near 39.79 kHz with the amplitude of 1.15 $V_p$ voltage. The result shows that the third-order IMD is around −31.08 dBc and the third-order intercept (TOI) is around 21.59 dBm.

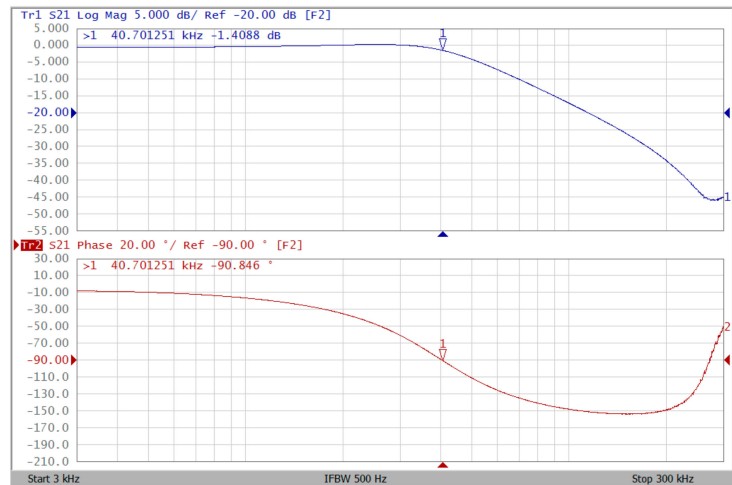

**Figure 18.** Measured gain and phase responses at $V_{o1}$ when $V_{i1} = V_{in}$ and $V_{i2} = 0$.

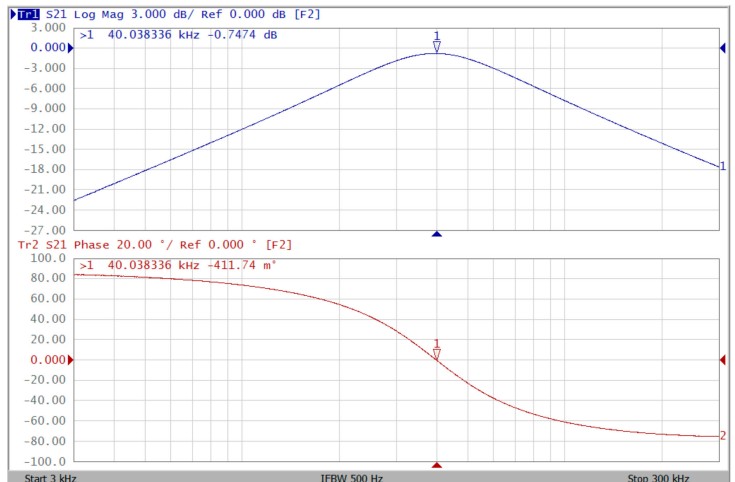

**Figure 19.** Measured gain and phase responses at $V_{o2}$ when $V_{i1} = V_{in}$ and $V_{i2} = 0$.

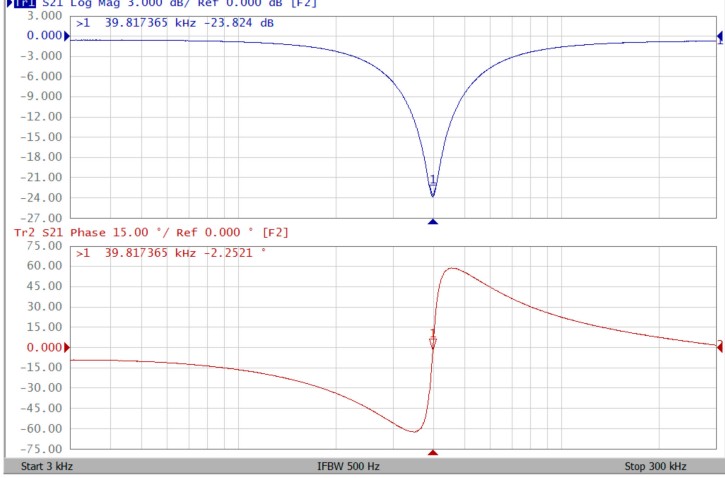

**Figure 20.** Measured gain and phase responses at $V_{o3}$ when $V_{i1} = V_{in}$ and $V_{i2} = 0$.

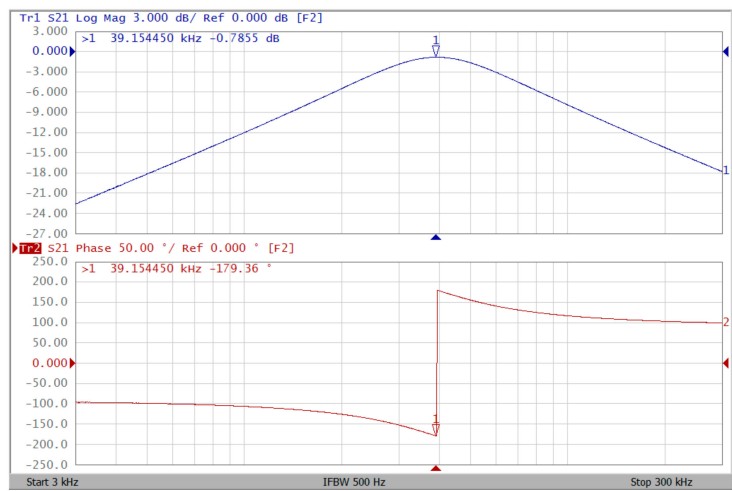

**Figure 21.** Measured gain and phase responses at $V_{o1}$ when $V_{i1} = 0$ and $V_{i2} = V_{in}$.

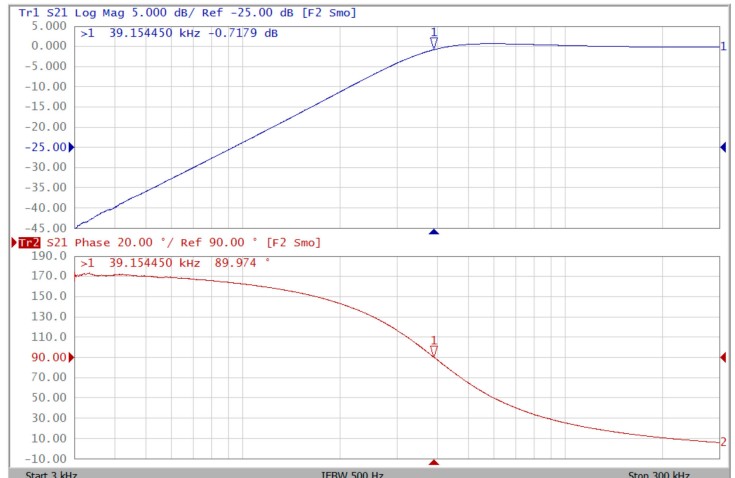

**Figure 22.** Measured gain and phase responses at $V_{o3}$ when $V_{i1} = 0$ and $V_{i2} = V_{in}$.

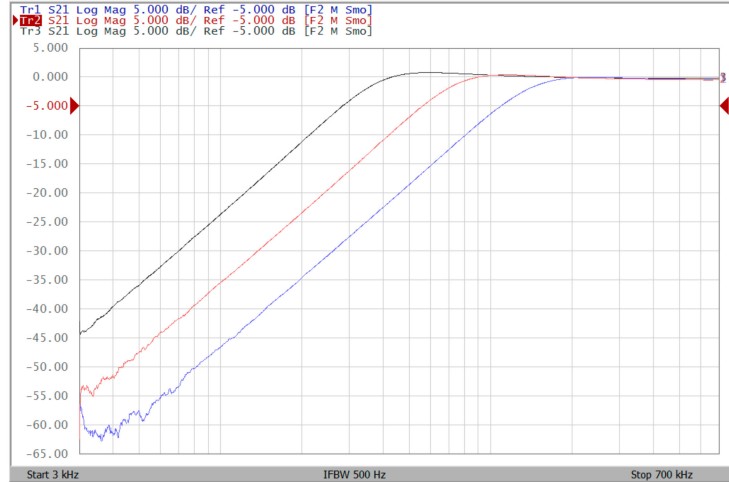

**Figure 23.** Measured gain responses of HP ($V_{o3}$) filters by varying $f_o$ while keeping Q ($f_o$ = 39.15 kHz—black line; $f_o$ = 76.34 kHz—red line; and $f_o$ = 142.21 kHz—blue line).

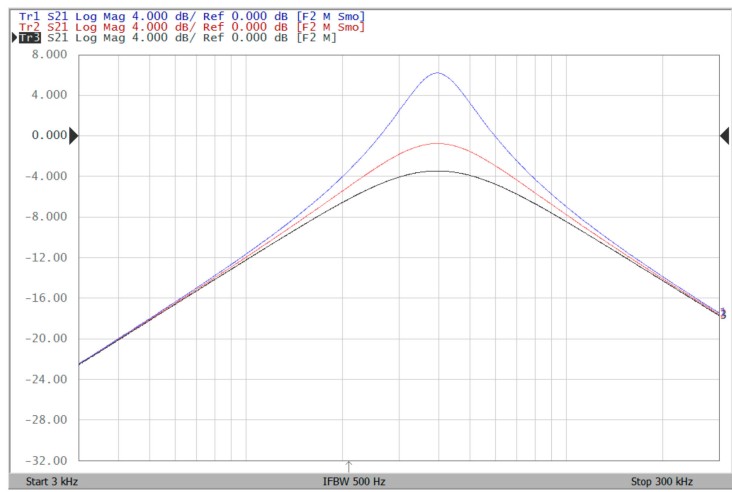

**Figure 24.** Measured gain responses of BP ($V_{o2}$) filters by varying Q while keeping $f_o$ (Q = 0.89—black line; Q = 1.06—red line; and Q = 2.12—blue line).

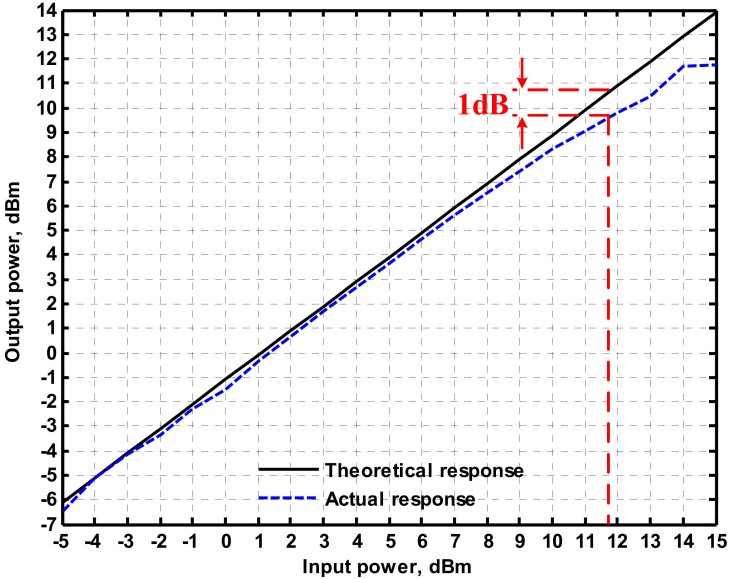

**Figure 25.** Measured P1dB of the IBP filter with input power at $V_{o1}$ when $V_{i1}$ = 0 and $V_{i2}$ = $V_{in}$.

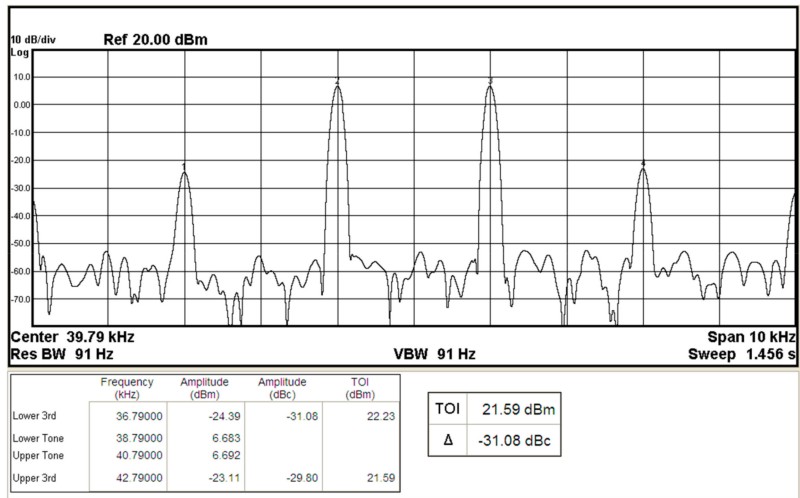

**Figure 26.** Measured IBP filter of the two-tone test with the amplitude of 1.15 $V_P$ voltage.

### 4.2. Voltage-Mode Quadrature Sinusoidal Oscillator

The proposed oscillator was simulated using OrCAD PSpice, and measured using the Agilent N9000A CXA signal analyzer. The supply voltages were $V_{DD} = -V_{SS} = 6$ V. The passive components were chosen as $C_1 = C_2 = 1$ nF and $R_1 = R_2 = R_3 = 4$ k$\Omega$. $R_4$ was adjusted to 3.98 k$\Omega$ to start the oscillations. The theoretical oscillation frequency using this design was 39.79 kHz. The steady state waveforms for both the quadrature voltages are shown in Figure 27. Figure 28 shows the quadrature relationship between the generated waveforms of $V_{o1}$ and $V_{o3}$ in the X–Y plane. From the simulation results, the oscillation frequency of $f_o = 38.84$ kHz was obtained, which agrees very well with the theoretical analysis The results of the $V_{o1}$ total harmonic distortion analysis are summarized in Table 3.

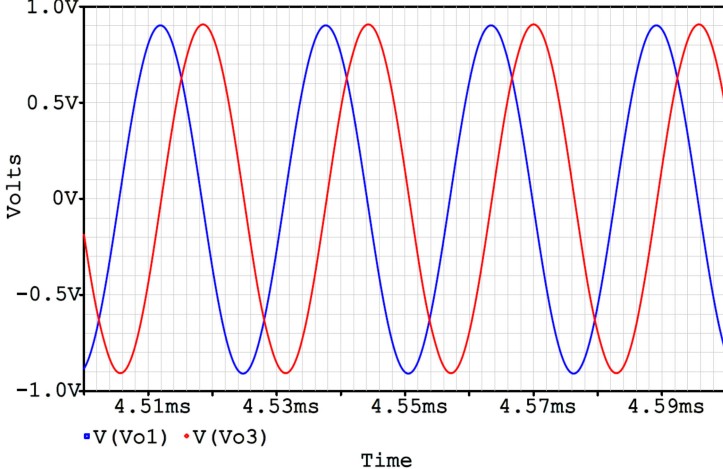

**Figure 27.** Simulated quadrature output waves $V_{o1}$ (blue) and $V_{o3}$ (red) of Figure 4.

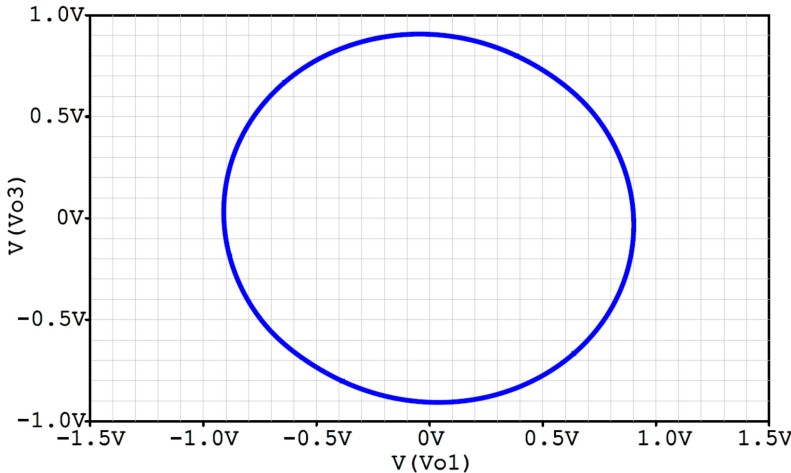

**Figure 28.** Simulated quadrature output waves $V_{o1}$ and $V_{o3}$ in the X–Y plane of Figure 27.

**Table 3.** Total harmonic distortion analysis of $V_{o1}$ in Figure 4.

| Harmonic Number | Frequency (Hz) | Fourier Component | Normalized Component | Phase (Degree) | Normalized Phase (Degree) |
|---|---|---|---|---|---|
| 1 | $3.98 \times 10^4$ | $8.94 \times 10^{-1}$ | 1.00 | $7.94 \times 10^1$ | 0 |
| 2 | $7.96 \times 10^4$ | $1.45 \times 10^{-2}$ | $1.62 \times 10^{-2}$ | $6.74 \times 10^1$ | $-9.14 \times 10^1$ |
| 3 | $1.19 \times 10^5$ | $7.03 \times 10^{-3}$ | $7.86 \times 10^{-3}$ | $6.90 \times 10^1$ | $-1.69 \times 10^2$ |
| 4 | $1.59 \times 10^5$ | $3.28 \times 10^{-3}$ | $3.66 \times 10^{-3}$ | $5.23 \times 10^1$ | $-2.65 \times 10^2$ |
| 5 | $1.99 \times 10^5$ | $2.49 \times 10^{-3}$ | $2.78 \times 10^{-3}$ | $4.31 \times 10^1$ | $-3.54 \times 10^2$ |
| 6 | $2.39 \times 10^5$ | $1.88 \times 10^{-3}$ | $2.11 \times 10^{-3}$ | $4.71 \times 10^1$ | $-4.29 \times 10^2$ |
| 7 | $2.79 \times 10^5$ | $1.38 \times 10^{-3}$ | $1.54 \times 10^{-3}$ | $3.14 \times 10^1$ | $-5.24 \times 10^2$ |
| 8 | $3.18 \times 10^5$ | $1.19 \times 10^{-3}$ | $1.33 \times 10^{-3}$ | $2.85 \times 10^1$ | $-6.07 \times 10^2$ |
| 9 | $3.58 \times 10^5$ | $9.40 \times 10^{-4}$ | $1.05 \times 10^{-3}$ | $3.52 \times 10^1$ | $-6.79 \times 10^2$ |
| DC component = $-2.560956 \times 10^{-2}$ | | | | | |
| Total harmonic distortion = 1.887184% | | | | | |

Regarding the experimental tests for the proposed oscillator, Figure 29 represents the oscilloscope output waveforms, $V_{o1}$ and $V_{o3}$, of the proposed oscillator in Figure 4. This is designed for the oscillation frequency of $f_o$ = 39.79 kHz with the component values of $C_1 = C_2$ = 1 nF and $R_1 = R_2 = R_3$ = 4 k$\Omega$, and the oscillation starts from $R_4$ = 3.77 k$\Omega$. The measured oscillation frequency, 38.55 kHz, is very close to theoretical value and the error is $-3.12\%$. Figure 30 shows the Lissajous pattern of the $V_{o1}$ and $V_{o3}$ outputs of the experimental result in Figure 29. Figure 31 shows the frequency spectrum of the voltage output, $V_{o1}$, of the oscillator. The magnitude of the second harmonic is 38.87 dB less than the magnitude of the fundamental harmonic. The measured oscillation frequency, 37.072 kHz, is close to the theoretical value. Obviously, the experimental results are consistent with the theoretical values. Figure 32 shows how phase noise was calculated using the Agilent phase noise measurement solution. The phase noise in the proposed oscillator is lower than $-85.7$ dBc/Hz (at 1 kHz offset). Obviously, the experimental results are consistent with the theoretical values.

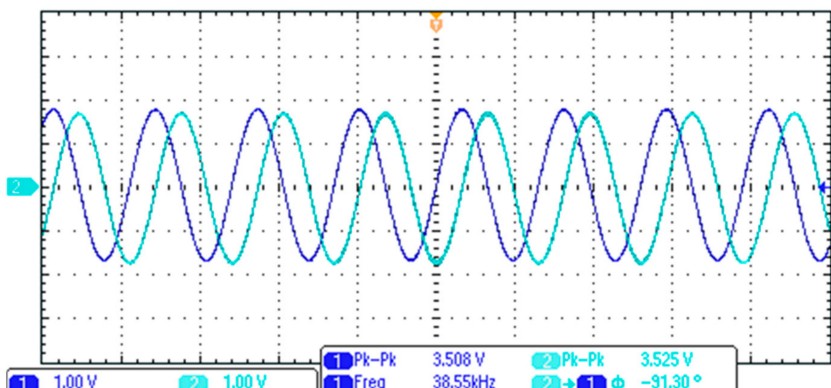

**Figure 29.** Experimental of the quadrature outputs time-domain results $V_{o1}$ (blue) and $V_{o3}$ (cyan).

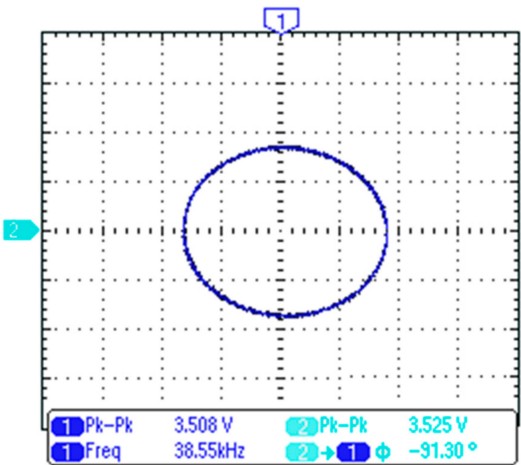

**Figure 30.** Lissajous pattern of $V_{o1}$ and $V_{o3}$ outputs in Figure 29 (horizontal scale: 1 V/div; vertical scale: 1 V/div).

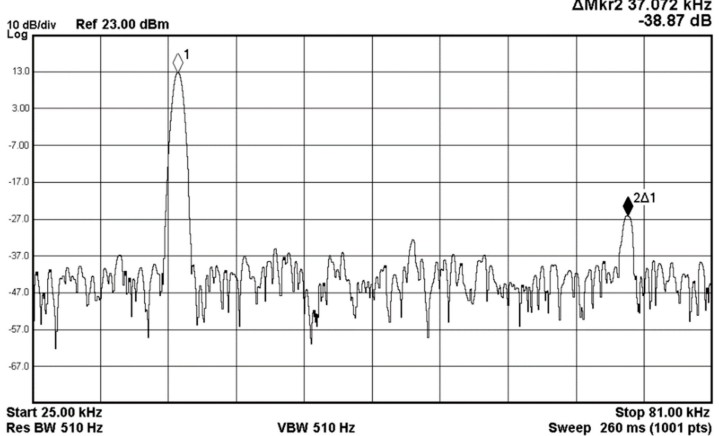

**Figure 31.** Measured frequency spectrum of $V_{o1}$.

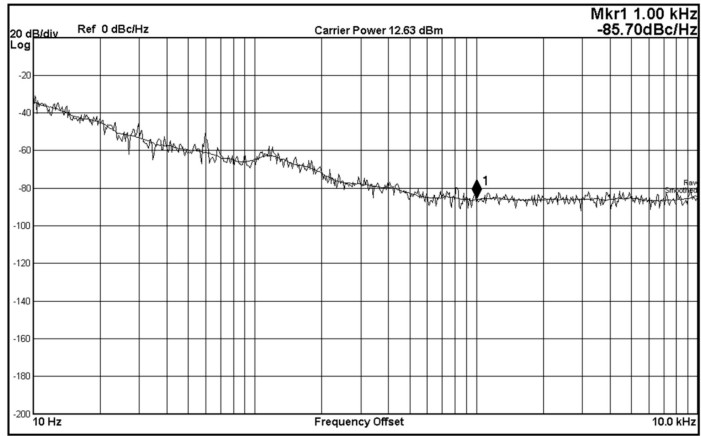

**Figure 32.** Measured phase noise of the proposed oscillator in Figure 4.

## 5. Conclusions

In this paper, a new two inputs and three outputs voltage-mode multifunction biquadratic filter with high-input impedances is presented. The proposed circuit uses three CFOAs, two grounded capacitors, and three resistors. All the X ports of the CFOAs in the proposed filter are connected directly to resistors. This design offers the feature of a direct incorporation of the parasitic resistance at the X

terminal of the CFOA as a part of the main resistance. The circuit offers the advantage of orthogonal control of $\omega_o$ and Q and simultaneously provides LP, BP, and BR responses while the IBP and HP responses can be easily obtained by applying another input voltage signal. Moreover, the filter can be easily converted into a quadrature oscillator, and the CO and the FO can be controlled independently. The experimental results validate that the functions of the proposed filter and oscillator are excellent.

**Author Contributions:** S.-F.W. and H.-P.C. conceived and designed the theoretical verifications; the optimization ideas were provided by Y.K.; H.-P.C. analyzed the results and wrote the paper; P.-Y.C. performed the simulations and experiments.

**Funding:** This research was funded by the Ming Chi University of Technology.

**Acknowledgments:** We are grateful to the editors and referees for their invaluable suggestions for improving the paper.

**Conflicts of Interest:** The authors declare no conflict of interest.

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
