# Peer review of "A CFOA-Based Voltage-Mode Multifunction Biquadratic Filter and a Quadrature Oscillator Using the CFOA-Based Biquadratic Filter"

_applsci, doi:10.3390/app9112304_

Round 1

Reviewer 1 Report

The authors propose a new topology for a biquadratic filter based on CFOA. The paper contains extensive calculations, simulations and measurement results of the possible circuit variations. 

I have only some minor remarks:

line 99: it must be input impedance, and terminal W does not appear in any Figure.

starting from Fig. 7, some writing in the figures is too small, see also Figs. 27 and 28

Author Response

REPLY TO REVIEWERS’ COMMENTS

The authors would like to express his gratitude to anonymous reviewers for carefully reviewing the paper, for many thoughtful comments in the original manuscript. The manuscript has been revised and improved according to the suggestions of reviewers. The changes in the revised manuscript are marked in yellow.

Comments and Suggestions for Authors

The authors propose a new topology for a biquadratic filter based on CFOA. The paper contains extensive calculations, simulations and measurement results of the possible circuit variations.

I have only some minor remarks:

line 99: it must be input impedance, and terminal W does not appear in any Figures

starting from Fig. 7, some writing in the figures is too small, see also Figs. 27 and 28.

Ans: Thanks for your comment. The X terminal is low input impedance and the O terminal is low output impedance. Terminal W in line 99 of original manuscript should be replaced by terminal O, and we have corrected it. We have also enlarged the font size in figures 14, 27 and 28. Please see line 99, line 234, line 290 and line 292 of the revised manuscript.

Reviewer 2 Report

Basically, it is well written paper. And I just recommend several things to strengthen this paper’s quality and to help reader’s understanding.

1.     Did you calculate or measure the total power consumption? If you include the information of voltage and current in detailed, it will be better.

2.     In Fig30, what are the units for x-axis and y-axis?

3.     In Fig.25.  There are two lines. If you denote which lines are the trend line and measured values, it will be better.

4.     I am little bit confused about line 242~250. You mentioned that 39.79KHz in line 245,248 but, you describe that you got 37.53kHz in line 247. Why did you get the different frequency. Because of the component? Or, did you set up the different frequency with intention?

5.    In Fig 13,you applied +/- 6V DC and then you got the voltage less 2.4Vpp. Which determine the factor? If you can describe it, can you increase the output voltage amplitude? And then, what is the trade-off for it? 

Author Response

REPLY TO REVIEWERS’ COMMENTS

The authors would like to express his gratitude to anonymous reviewers for carefully reviewing the paper, for many thoughtful comments in the original manuscript. The manuscript has been revised and improved according to the suggestions of reviewers. The changes in the revised manuscript are marked in yellow.

Reviewer report form #2

Comments and Suggestions for Authors

Basically, it is well written paper. And I just recommend several things to strengthen this paper’s quality and to help reader’s understanding.

1. Did you calculate or measure the total power consumption? If you include the information of voltage and current in detailed, it will be better.

Ans: Thanks for your comment. We have revised line 244 and line 245, page 11 of the manuscript. The power consumption is approximately 180 mW under ±6 V of supply voltages and 15 mA constant output current.

2. In Fig30, what are the units for x-axis and y-axis?

Ans: Thanks for your comment. The horizontal scale is 1 V/div and the vertical scale is 1 V/div. We have revised line 310, page 17 of the manuscript.

3. In Fig.25. There are two lines. If you denote which lines are the trend line and measured values, it will be better.

Ans: Thanks for your comment. We have denoted the trend line (theoretical response) and measured value (Actual response) in Fig. 25. Please see line 275, page 14 of the manuscript.

4. I am little bit confused about line 242~250. You mentioned that 39.79KHz in line 245,248 but, you describe that you got 37.53kHz in line 247. Why did you get the different frequency. Because of the component? Or, did you set up the different frequency with intention?

Ans: Thanks for your comment. In Figures 18 to 22, the theoretical quality factor Q = 1 and theoretical angular frequency fo = 39.79 kHz, when passive elements are set as C1 = C2 = 1 nF and R1 = R2 = R3 = 4 kΩ. The power consumption is approximately 180 mW under ±6 V of supply voltages and 15 mA constant output current. In Figure 23, the measured values of angular frequency are equal to 39.15 kHz, 76.34 kHz, and 142.21 kHz, respectively, by adopting C1 = C2 = 1 nF and equal resistors of R1 = R2 = R3 with 4 kΩ, 2 kΩ, 1 kΩ. We have revised from line 242 to line 247 of the manuscript.

5. In Fig 14, you applied +/- 6V DC and then you got the voltage less 2.4Vpp. Which determine the factor? If you can describe it, can you increase the output voltage amplitude? And then, what is the trade-off for it?

Ans: Thanks for your comment. Figure 14 shows an example with repeating sinusoidal input signal of 39.79 kHz frequency. In Figure 14, the percentage of total harmonic distortion is 0.6%, when the sinusoidal input signal of an amplitude 1 Vp at 39.79 kHz is applied. Figure 15 shows the dependence of the IBP filter output harmonic distortion on the input signal amplitude. As can be seen in Figure 15, when the input signal amplitude increases by 2 Vp, its total harmonic distortion will increase by 3.2%. We have revised from line 205 to line 207 of the manuscript.
